# Mesonephric-Like Adenocarcinomas a Rare Tumor: The Importance of Diagnosis

**DOI:** 10.3390/ijerph192114451

**Published:** 2022-11-04

**Authors:** Stefano Restaino, Giulia Pellecchia, Angelica Tulisso, Chiara Paglietti, Maria Orsaria, Claudia Andreetta, Elena Poletto, Martina Arcieri, Monica della Martina, Anna Biasioli, Laura Mariuzzi, Lorenza Driul, Giovanni Scambia, Giuseppe Vizzielli

**Affiliations:** 1Department of Medical Area (DAME), Clinic of Obstretics and Gynecology “Santa Maria della Misericordia”, University Hospital Azienda Sanitaria Universitaria Friuli Centrale, University of Udine, 33100 Udine, Italy; 2Department of Medicine, Pathological Anatomy Section, University of Udine, 33100 Udine, Italy; 3Institute of Pathological Anatomy, Azienda Sanitaria Universitaria Friuli Centrale (ASUFC), Presidio Ospedaliero “Santa Maria della Misericordia”, 33100 Udine, Italy; 4Department of Medical Oncology, Azienda Sanitaria Universitaria Friuli Centrale (ASUFC), Ospedale S. Maria della Misericordia, 33100 Udine, Italy; 5Medical Area Department, Institute of Pathological Anatomy, Director School of Specialisation in Pathological Anatomy, University of Udine, Azienda Sanitaria Universitaria Friuli Centrale, p.le S.Maria della Misericordia, 33100 Udine, Italy; 6Dipartimento per le Scienze Della Salute Della Donna, del Bambino e di Sanità Pubblica, UOC Ginecologia Oncologica, Fondazione Policlinico Universitario Agostino Gemelli IRCCS, 00168 Roma, Italy; 7Faculty of Medicine and Surgery, Università Cattolica del Sacro Cuore, 00168 Roma, Italy

**Keywords:** mesonephric-like adenocarcinomas, confounding hystology, the importance of diagnosis, case report

## Abstract

Mesonephric-like adenocarcinomas (MLA) are rare neoplasms that arise in the uterine body and ovary and have been added to the World Health Organisation’s recent 2020 classification of female genital cancers. The pathogenesis of MLA is unknown and it remains debated whether they represent mesonephric carcinomas (Wolffian) arising in the endometrium/ovary or endometrioid carcinomas (Müllerian) closely mimicking mesonephric carcinomas. Here we report the case of a 57-year-old woman with an initial misdiagnosis of endometrioid adenocarcinoma on diagnostic biopsy. The patient came to our clinical evaluation for the appearance of menometrorrhagia complicated by anemia for several months. Therefore, she underwent pelvic echo-flowmetry, with indication for diagnostic hysteroscopy with endometrial biopsy, which yielded a positive result for endometrioid endometrial adenocarcinoma. Following staging CT scan and targeted examinations on pulmonary findings, the patient underwent surgery with surprise of definitive diagnosis deponent for endometrial MLA. Our intention is to establish a brief review of the scientific evidence in the literature and the tools available for a correct histological diagnosis, in the light of the scant anatomopathological evidence. Our question gives rise to the motive for the publication: is immunohistochemistry the right way to resolve the diagnostic error at histology, which is usually the only source of diagnostic certainty? This case is intended to alert of diagnostic error that risked having the patient treated as a neoplasm with a favorable prognosis and low degree of aggressiveness instead of for a very aggressive and poor prognosis tumor such as MLA.

## 1. Introduction

Mesonephric-like adenocarcinomas (MLA) are rare neoplasms arising in the uterine corpus and ovary which have been added to the recent 2020 World Health Organization Classification of Female Genital Tumors [1]. It is a recently recognized adenocarcinoma of the uterine corpus and ovary characterized in 2016 by McFarland et al. [2] in a series of 12 cases. MLA of the uterine corpus has considerable morphologic and immunohistochemical overlap with conventional mesonephric adenocarcinoma, but is differentiated from conventional mesonephric adenocarcinoma, a tumor of wolffian origin, by consistently originating/involving the endometrium, lacking associated mesonephric remnants or hyperplasia and infrequently involving the uterine cervix [3,4]. The pathogenesis of MLAs is unknown, and it remains debated whether they represent mesonephric (Wolffian) carcinomas arising in the endometrium/ovary or endometrioid (Müllerian) carcinomas that closely mimic mesonephric carcinomas [5]. MLAs represent 1% of endometrial carcinomas with a median age of onset around 52 years on an atrophic endometrium. The most frequent clinical presentation is atypical vaginal bleeding with transvaginal ultrasound finding of large exophytic-appearing neoformations >4 cm [6]. In a case series, Pors et al. reported as initial symptoms of MLA, although of markedly low incidence, either an incidental diagnosis, or uterine prolapse, during diagnostic investigations for infertility, or patients with uterine fibroids. MLA were initially correctly diagnosed on biopsy in 32% of cases [7]. The role of molecular analysis in cancer diagnostics is becoming increasingly important. In this regard, molecular analisys suggests that mesonephric-like carcinoma is characterized by *KRAS* missense mutation and an aggressive clinical behaviour [8,9].

Due to its rarity and propensity to have a variety of architectural patterns mimicking more commonly encountered tumours, the diagnosis can be a challenge. Here we report a case of a 57-years-old woman with advanced-stage disease, and discuss the clinical characteristics, pathological diagnosis and treatment of this tumour. Taking our cue from the clinical case: the rarity of the pathology and the diagnostic error at initial histology; we would like to dissect in light of the literature to date, what is known and what tools we have to delineate this tumor pathological entity. The patient obviously gave informed consent to the publication of the clinical case.

## 2. Case Presentation

The patient came to our clinical evaluation for the appearance of menometrorrhagia complicated by anemia for several months. In remote pathological history, no comorbidities; a previous operative hysteroscopy in 2008 with endometrial polypectomy, whose histological examination was negative, was reported. The patient had not performed pap smears for several years. At pelvic ultrasound examination, an endometrial cavity was entirely occupied by an inhomogeneous and richly vascularized color score (CS) 4 multilocular formation of 25 mm. A diagnostic hysteroscopy with endometrial biopsies was performed. Histological examination revealed almost exclusive neoplastic cells with low-grade features, organized in glandular and papillary features, and a final diagnosis of endometrial adenocarcinoma of FIGO2 grade endometrioid histotype was done. The patient, therefore, underwent a preoperative staging CT scan that revealed a suspicious lung lesion. Therefore, after a multidisciplinary clinical consultation with radiologist, pathologist and oncologist colleagues, it was decided to perform a bronchoscopy with a CT-guided biopsy of the lung lesion. It was negative for malignancy. So, the patient underwent laparoscopy/xipho-pubic laparotomy surgery with bilateral hysteroannessiectomy, vaginal biopsy, omentectomy, intercavoaortic and paracaval lymphadenectomy, bilateral pelvic lymphadenectomy, and removal of para-aortic lymph nodes with residual tumor (RT) of 0. This is because upon intraoperative findings, on palpation, they were appreciated as bulky lymph nodes. Already at preoperative staging CT scan, enlarged lymph nodes were reported in suspected relation to productive process near the common iliac vessels bilaterally and left para-aortic. The postoperative course was however smooth, and free of complications, and the patient was discharged home 5 days after surgery.

## 3. Pathological Evaluation

A gross examination of the uterus revealed a polypoid uterine mass of 6 cm, which infiltrated the inner half of the myometrium, extended towards the cervix, and infiltrated the vagina. All lymph node stations had some macroscopically metastatic lymph nodes. Bilateral adnexa and omentum were grossly normal. Microscopically the tumor was composed of varied growth patterns including tubulo-cystic, papillary, glandular, solid, and glomeruloid areas. In some cases, there was marked dilatation of some of the tubules with attenuation of the epithelial lining. The epithelial cells are cuboidal or flat, and exhibited moderate nuclear atypia, clear or vesicular and angulated nuclei, and small nucleoli. The cytoplasm was generally scant. No squamous metaplasia was identified. Focally there were neoplastic cells with increased nuclear atypia, mimicking serous carcinoma. Intraluminal eosinophil secretions were rarely recognized. Mitotic activity was variable. The mass infiltrated the inner half of the myometrium, the vagina, and the endocervical glands without involving the cervical stroma. There was an extensive lymphovascular invasion and seventeen lymph nodes, both pelvic, para-aortic, and para-caval were positive for macrometastasis with extracapsular extension. The definitive histological examination completely overturned the initial histological diagnosis: mesonephric-like endometrial adenocarcinoma with stage pT3bN2a according to the TNM classification of the Union for International Cancer Control (UICC), 8th edition. Immunohistochemically, tumor cells showed diffusely positive staining for PAX8 and GATA3, p53 wild type staining, patchy positivity for TTF-1, Napsin A, Estrogen receptor, retained staining for the mismatch repair proteins (MLH-1, PMS-2, MSH-2, MSH-6) and negative staining for Progesterone receptor. Genetic analysis identified *KRAS* mutation at codon 12 (G12C alteration). Neither *POLE* nor p53 mutations were detected.

In view of the rarity of the tumor histotype and according to the guidelines of the major scientific societies of gynecological oncology and the judgment of the multidisciplinary team of the gynecological-oncological pathway, the disease was treated as a high grade and worse prognosis. The patient has therefore started an adjuvant treatment with chemotherapy based on carboplatin and paclitaxel for six cycles. She was enrolled in an experimental oncology clinical trial involving dostarlimab vs. placebo. From the second chemotherapy cycle, unfortunately, paclitaxel administration was discontinued due to G3 allergic reaction (onset of lumbar pain NRS 10 and diffuse heat; no signs of bronchospasm, but symptomatic hypotension without loss of consciousness; episode of vomiting and desaturation). Otherwise, gyneco-oncologic follow-up, at the time of writing, is negative. She recently underwent a follow-up abdominal CT scan: lesions previously found in the lung site are stable in size and shape.

## 4. Discussion

Mesonephric carcinoma is a distinct and rare histologic subtype of carcinoma, mainly arising in the uterine cervix in the background of normal or hyperplastic mesonephric remnants, representing <1% of all carcinomas at this site. Patients most commonly present with abnormal vaginal bleeding. Histologically, it is characterized by a variety of architectural patterns, including tubular, ductal/glandular, papillary, solid, and retiform. Intraluminal eosinophil secretion can be frequently observed. Neoplastic cells may be cuboidal or columnar, with scant cytoplasm [4,10]. The nuclei often have irregular nuclear membranes, nuclear grooves, and pseudoinclusions; features reminiscent of papillary thyroid carcinoma [4,10].

Recently, carcinoma have been described arising in the uterine corpus and ovary which share morphologic features with cervical mesonephric carcinoma but were not associated with mesonephric remnants/hyperplasia and were termed mesonephric-like adenocarcinoma [2,11].

In their study, Pors et al. analyzed the clinical, histopathological, and prognostic features of MA and MLA of cervix one and endometrium and ovary the other, respectively [7]. They found that only a small number of MA and MLA as a whole were correctly diagnosed by pathologists on biopsy. Of this aspect that recalls our case, emphasizing the need to study and implement scientific evidence on the subject, the authors give several possible explanations, including the uncommon nature of these neoplasms, their variety of architectural patterns mimicking other neoplasms, and the fact that MLA is relatively recently described neoplasm and may not be known to some pathologists. Alternatively, some authors have hypothesized that MLA derives from Mullerian, rather than mesonephric, structures and differentiates along mesonephric lines; this hypothesis resulted in the term MLA [2,7]. If MLA is actually of Mullerian origin, with subsequent mesonephric transdifferentiation, it is plausible that MLA shares a common precursor with other Mullerian neoplasms, such as endometrial atypical hyperplasia. Unlike MA of the cervix, MLA of the endometrium appear to arise from the endometrium rather than the myometrium, where residual mesonephric remnants would theoretically reside, and have not been found to be associated with mesonephric remnants [2,7]. In addition, several cases of MLA, especially in the ovary but also in the endometrium, have been shown to coexist with prototypical Mullerian neoplasms, further suggesting the possibility of Mullerian origin in at least a subset of these tumors [7]. The authors analyzed 99 MA and MLA cases and concluded they share an aggressive behavior, difficulty to be detected, presentation often at an advanced stage with a high frequency of distant recurrences, in above all marked propensity for pulmonary recurrences.

The term mesonephric-like adenocarcinoma is used because while it closely resembles mesonephric adenocarcinoma morphologically and immunohistochemically, there are other features suggestive of Mullerian origin, including that in the uterine corpus these neoplasms appear to arise from the endometrium. In addition, some tumors are confined to the endometrium without deep myometrial involvement, where mesonephric remnants would theoretically exist. Histogenesis of these tumors is under debate whether these represent true mesonephric carcinoma that arises in the uterine corpus and ovary or Mullerian (probably endometrioid) carcinomas that closely mimic mesonephric adenocarcinoma [12].

Molecular alterations most frequently encountered in this neoplasia are characterized by KRAS mutation at codon 12, as well as a high frequency of gains of chromosomes 1q, 10, 12 and loss of chromosome 1p. KRAS activating mutations are the most common molecular alteration detected in mesonephric carcinoma which results in constitutive activation of mitogen-activated protein kinase (MAPK), with subsequent activation of a variety of downstream targets. These latter lead to the expression of genes involved in proliferation, differentiation, and survival [13].

Mesonephric-like carcinoma has an aggressive clinical course, and a higher rate of recurrence and distance metastasis, compared to the more common endometroid and serous-type carcinomas [10].

Our interest in the publication of this clinical case can be easily deduced from how limited the knowledge in the literature on the subject is. It is a tumor histotype of recent identification with considerable limits on the current definition of clinical pathological and prognostic characteristics [7]. In fact, only 119 cases of uterine and 39 ovarian MLA carcinoma have been reported and they are all extrapolated from case reports/series [10]. Another peculiar feature of the examined case and that recalls the literature concerns the diagnostic difficulty from the anatomo-pathological point of view: an element of great surprise if we consider that any diagnosis of certainty is determined histologically. The first histological diagnosis based on our patient’s endometrial biopsy suggested an endometrioid histotype, while the final examination of the surgically excised tissues radically changed the diagnosis for the worse (stage, prognosis, and therapy required) [10]. A possible reason for this behavior is highlighted by pathologists: such tumors often have multiple cellular growth patterns (glandular, tubular). Hence the need to implement our scientific knowledge on the subject and especially to increase the degree of scientific evidence. Another noteworthy element is the frequent site of pulmonary secondarisms [14]: our patient underwent needle aspiration of a suspected pulmonary lesion.

## 5. Conclusions

There are no guidelines in the literature on the gynecological, and oncological management of these neoplasms. To date, therefore, we cannot make use of international guidelines for the management of these patients as there is still much to be defined about this neoplastic entity before standardizing its treatment. Therefore, a multidisciplinary team, possibly composed of pathologists who are experts on the subject and who have already published scientific papers on the subject, as other authors have already pointed out, would be desirable. Immunohistochemistry in fact in this case as in others is not entirely specific and only a combination of markers can support a diagnosis, based, however, primarily on morphology and aggressive clinic in a carcinoma that morphologically makes glands so paradoxically, it is a low grade. So, we have shown how histologic diagnosis is insidious and we often run into diagnostic errors: errors in a diagnosis that should be one of certainty, as only histology can be. Even, it has been debated whether MLA represents a distinct clinical entity or is a subtype of Mullerian carcinomas, such as endometrioid adenocarcinoma. In light of this, which is absolutely central as an object of study on the topic, an additional doubt emerges about the case we present: was it a diagnostic error, or rather will we find that endometrial MLA and endometrioid histotype have a common matrix?

The limitation of our work as well as others already in the literature is that there is not a large case series to study; due to the rarity and novelty especially of this neoplastic entity. Thus, as it often happens, we advance in the form of literature reviews and case reports to meanwhile provide those working in the field with the best quality of information possible, in order to better manage patients. It is therefore desirable that clinical studies be put in place on several fronts: from the genetics of the tumor to its immunopathological features, immunohistochemistry, and last but not least clinical, treatment and prognosis.

## Data Availability

The data that support the findings of this study are available from the corresponding author, upon request.

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
