# Peer review of "Mesonephric-Like Adenocarcinomas a Rare Tumor: The Importance of Diagnosis"

_ijerph, 2022, doi:10.3390/ijerph192114451_

Round 1
Reviewer 1 Report
The manuscript entitled "Mesonephric-like adenocarcinomas a rare tumor: the importance of diagnosis" is a brief review of mesonephric adenocarcinomas and the tools available for a correct diagnosis. Additionally, the authors question if immunohistochemistry can solve the diagnostic error in histology.
The presented information in the manuscript can be used in further studies.
Some points to clarify to increase the visibility of the manuscript.
1 - Please, clarify the reason only one case was reported.
2 - Discussion section and conclusion.
The authors cite some numbers regarding the cases reported. Perhaps, please, comparing with similar carcinomas and increasing the different numbers regarding the disease and comparing, will increase the visibility of the manuscript.
3 - Please, the conclusion section can be improved significantly. The perspectives are very shallow.
Author Response
Dear reviewer, we thank you for your review, comments and requested clarifications.
The case report as you pointed out, points to the importance of diagnostic aids of certainty supporting an already certainty diagnosis such as anatomo-pathological. This is a single case to emphasize precisely the initial misdiagnosis found.
- Please, clarify the reason only one case was reported:
The work that we have submitted to you is inherent to the single clinical case that we had in our center: as the albeit little literature shows us, this is a very rare neoplastic pathology for which the case history is rather limited. Moreover, our intent was to emphasize the risk of diagnostic mistake that it happens to incur, although it is not a rule.
- The authors cite some numbers regarding the cases reported. Perhaps, please, comparing with similar carcinomas and increasing the different numbers regarding the disease and comparing, will increase the visibility of the manuscript.
Building on the work of Pors et al., we implemented the discussion by adducing the data reported in this study, which is probably the largest on the subject to date. We thank you for the suggestion.
- Please, the conclusion section can be improved significantly. The perspectives are very shallow.
Thank you for your comment; we have taken care to better explicate some of the concepts we wanted to convey and leave as a starting point for future studies to come.
Best regards,
The authors
Reviewer 2 Report
Feedback using CARE Checklist (https://www.care-statement.org/):
Item 2) Missing "case report" in Keywords
Item 3a) Abstract: What is unique about this case? e.g. not already covered in the Deolet et al 2021 paper (PMID: 33670088)
Item 3b) Abstract: Missing main symptoms from patient
Item 3c) Abstract: Missing therapeutic interventions and outcomes
Item 3d) Abstract: What is the main “take-away” lesson(s) from this case?
Item 4) Introduction: missing summary why this case is unique (see item 3a above)
Item 5c) Patient information missing patient medical history
Item 7) no timeline
Item 10c) Line 126: how was paclitaxel allergic reaction assessed
Item 11a) Missing a scientific discussion of the strengths AND limitations associated with this case report
Item 11c) missing scientific rationale for any conclusions
Item 11d) What was the primary “take-away” lessons of this case report? Please be clearer in conclusions. Was immunohistochemistry 100% necessary for the correct diagnosis after the surgical excised tissues were analysed by gross examination and standard histology?
Item 12) patient perspective missing
Comments
Lines 106, 116: references to "Fig.1" and "Fig.2", but no figures found in paper pdf provided, no links found on mdpi website.
Suggestions
Line 31: Suggest change "Mesonephric adenocarcinomas (MLA)" to "Mesonephric-like adenocarcinomas (MLA)"
Line 64: Suggest change "As we know, the role of molecular analysis in cancer diagnostics is increasingly preponderant." to "The role of molecular analysis in cancer diagnostics is becoming increasingly important."
Line 74: Delete extra "The"
Lines 132, 133, 135, 136: need citation(s). Citations should be in text, not simply at end of paragraph.
Line 143: Suggest change "This term is used although this tumour..." to "The term mesonephric-like adenocarcinoma is used because while it..."
Line 170: Suggest change "The first histological diagnosis, on our patient's endometrial biopsy, deposed in fact for an endometrioid histotype, differently from the final examination on surgical piece, radically changing for the worse stage, prognosis and type of therapy for the patient [15]." to
"The first histological diagnosis based on our patient's endometrial biopsy suggested an endometrioid histotype, while final examination of the surgical excised tissues radically changed the diagnosis for the worse (stage, prognosis and therapy required) [15]."
Author Response
Dear reviewer,
We thank you for your review. Below and in the text of the manuscript highlighted in red are the responses to your comments.
What is unique about this case report.
To bring out the limitations of knowledge by taking as an example a case of diagnostic mistake that risked having the patient treated as a neoplasm with a favorable prognosis and low degree of aggressiveness instead of for a very aggressive and poor prognosis tumor such as MLA.
Item 2) Missing "case report" in Keywords
Item 3a) Abstract: What is unique about this case? e.g. not already covered in the Deolet et al 2021 paper (PMID: 33670088)
Item 3b) Abstract: Missing main symptoms from patient
Item 3c) Abstract: Missing therapeutic interventions and outcomes
Item 3d) Abstract: What is the main “take-away” lesson(s) from this case?
Item 4) Introduction: missing summary why this case is unique (see item 3a above)
Item 5c) Patient information missing patient medical history
In accordance with the reviewers' comments and suggestions, we have modified these in the text.
Item 7) no timeline
There is no clear timeline as it is not the focus of our work, which aims to eviscerate tools to aid diagnosis, to prevent errors as in this case.
Item 10c) Line 126: how was paclitaxel allergic reaction assessed.
This was a type G3 infusion allergic reaction to the first cycle of paclitaxel: after 27.4 ml from the start of the infusion, the patient reported onset of lumbar pain NRS 10 and diffuse heat; there were no signs of bronchospasm, but symptomatic hypotension without loss of consciousness; episode of vomiting and desaturation. We specified it in the text.
Item 11c) missing scientific rationale for any conclusions.
Thank you for the suggestion. We have implemented the conclusions, clarifying our take home messages.
Item 11d) What was the primary “take-away” lessons of this case report? Please be clearer in conclusions. Was immunohistochemistry 100% necessary for the correct diagnosis after the surgical excised tissues were analysed by gross examination and standard histology?
The take-home message with this clinical case is that as of today, we do not have secure diagnostic tools to make this kind of diagnosis, mainly because the pathogenesis and genetics of this tumor are not yet known. Immunohistochemistry as is often the case, is not entirely specific. Only a combination of markers can support this diagnosis, based, however, primarily on morphology and aggressive clinic in a carcinoma that morphologically makes glands so would represent a low grade.
Item 11a) Missing a scientific discussion of the strengths AND limitations associated with this case report
Item 11c) missing scientific rationale for any conclusions
Thanks for the advice. We have tried to make our take home messages more explicit in the Conclusion section.
Item 12) patient perspectives missing.
Our patient is performing adjuvant maintenance therapy as per msintenance’ protocol. She performs
regular serious gynecological and radiological follow-ups every 3-4 months: so far they are negative.
We applied all your suggestions: thank you very much for your attention in reading our work.
Kind regards,
The authors.
Round 2
Reviewer 2 Report
I thank the authors for their edits which greatly improves this case-report.
Some of the English is a little difficult to read and could do with an English editor giving it a once over, but the understanding is there.
While I still have some doubt regarding the over-all novelty of this case, I'm satisfied there may be some benefit in its publication.